# Characterization of Bovine Papillomavirus Types Detected in Cattle Rumen Tissues from Amazon Region, Brazil

**DOI:** 10.3390/ani14152262

**Published:** 2024-08-04

**Authors:** Paulo Henrique Gilio Gasparotto, Igor Ribeiro dos Santos, Jerônimo Viera Dantas Filho, Mariana Soares da Silva, Fernanda dos Anjos Souza, Jennefer Caroline de Macedo Sousa, David Driemeier, Cláudio Wageck Canal, Flavio Roberto Chaves da Silva, Cíntia Daudt

**Affiliations:** 1Laboratório de Virologia e Parasitologia, Universidade Federal do Acre (UFAC), Rio Branco 69920-900, Brazilflavio.silva@ufac.br (F.R.C.d.S.); cintia.daudt@ufac.br (C.D.); 2Setor de Patologia Veterinária, Universidade Federal do Rio Grande do Sul (UFRGS), Porto Alegre 91540-000, Brazil; 3Programa de Pós-Graduação em Ciências Ambientais, Universidade Federal de Rondônia (UNIR), Rolim de Moura 76801-058, Brazil; 4Department de Medicina Veterinária, Centro Universitário São Lucas Ji-Paraná (UniSL Afya), Ji-Paraná 76907-524, Brazil; 5Laboratório de Microbiologia Molecular, Universidade Feevale, Novo Hamburgo 93525-075, Brazil; 6Laboratório de Virologia Veterinária, Faculdade de Veterinária, Universidade Federal do Rio Grande do Sul (UFRGS), Porto Alegre 91540-000, Brazil

**Keywords:** BPV, papilloma, phylogeny, cattle farming

## Abstract

**Simple Summary:**

Bovine papillomavirus (BPV) infection of the gastrointestinal tract (GIT) can induce the development of masses with high growth, resulting in breathing and eating obstructions leading to animal suffering and death. Beyond this, BPV is related to economic losses worldwide by depressing meat and milk production as well as cattle by-products. Using PCR followed by Sanger sequencing, we were able to identify the high-risk Delta BPVs and the BPV44 on rumen cattle samples collected in slaughterhouses. These results can contribute to future epidemiological studies and vaccine studies regarding BPV infections.

**Abstract:**

The *Bos Taurus Papillomavirus*, commonly known as bovine papillomavirus (BPV), can cause lesions in the mucosa of the gastrointestinal tract (GIT) in cattle and induce the formation of papillomas in organs such as the pharynx, esophagus, rumen and reticulum. GIT papillomas can lead to feeding and breathing distress. Moreover, the sample collection is challenging, which reduces the BPV diagnosis in these organs. BPV can cause exophytic nodular, cauliflower-like, flat, filiform or atypical-shape papillomas at the epidermis. Histologically, the papillomas demonstrate orthokeratotic/parakeratotic hyperkeratosis and koilocytosis and, currently, BPV comprises 45 described types. The aim of this study was to carry out the genetic characterization of BPV present in rumen neoplastic lesions of cattle raised extensively in the Western Amazon region, Brazil. A total of 100 papillomatous ruminal samples were collected from animals slaughtered in Ji-Paraná and Urupá municipalities from the Rondônia state, Brazil. The samples were submitted to PCR using the primer pair FAP59/FAP64 and sequenced by the Sanger method. Histopathological analysis was performed on 24 samples, which had enough material for this purpose. As a result, samples were histologically classified as fibropapilloma and squamous papilloma. Among the samples analyzed, it was possible to identify the BPVs 2, 13 (*Delta* PVs) and 44, with one sample classified as a putative new subtype of BPV44. The present study could identify BPV13 and 44 types in cattle rumen tissues from the Brazilian Amazon region for the first time.

## 1. Introduction

Papillomaviruses (PVs) are a large and diverse group of double-stranded circular DNA viruses about 55-60 nm in diameter, lacking lipoprotein envelope and infecting a wide variety of animal species from fish to humans [1,2,3]. Bovine papillomavirus (BPV) induces papillomas and fibropapillomas in both mucosal and cutaneous epithelium, causing vulvovaginal, penile and ocular lesions as well as udder and teats papillomas, urinary bladder and gastrointestinal tract (GIT) cancers [4,5,6,7,8]. Currently, BPVs comprise 44 types classified into five genera (Delta, Xi, Epsilon, Dyokappa and Dyoxipapillomavirus). In addition, the BPV types 19, 21, 27 and tick-associated BPV are still not attributed to a genus [5,9,10,11]. PVs are classified based on the nucleotide sequence of the L1 gene, which differs between each PV genotype [9].

BPV1 and 2 are the most described BPV types in cattle worldwide but are commonly found infecting equids, giraffes, deer and other animal species [12]. In cattle, BPV1 and 2 have been described in different clinical specimens [13,14,15], being mainly associated with skin lesions [12,16,17] and urinary bladder cancer [7,8,12,18,19]. Infection of the gastrointestinal tract (GIT) of cattle by BPV induces the formation of lesions that affect organs such as the pharynx, esophagus, rumen and reticulum [20]. Small, nodular or thin papillomas associated with BPV1, 2 and 5 have already been described in the rumen mucosa [21]. Additionally, GIT papillomas have been primarily associated with BPV4, which can progress to carcinomas [22]. These lesions can lead to clinical signs such as rumen intermittent swelling, wheezing and drooling [22], resulting in feeding and breath difficulties [23]. In cattle and buffaloes, papillomas have been described in the rumen mucosa, ranging from high growth to small nodular, spherical and pedunculated lesions [21,23].

Bovine GIT papillomas are predominantly accidental findings at necropsy or in slaughterhouses. These lesions can progress to squamous cell carcinoma (SCC) frequently associated to the chronic consumption of *Pteridium* spp. (Dennstaedtiaceae) [8,24]. Benign tumors usually show spontaneous regression; however, they can remain and, in the presence of environmental and/or genetic cofactors, evolve into a malignant lesion [8,24,25]. Despite being a relevant etiological agent in cattle farming, the detection and characterization studies of PV in animals are still deficient, especially regarding cattle rumen lesions [26,27]. BPV infection of the GIT can extend from the mouth, tongue, rumen and reticulum and can present high mass growth, which can result in breathing and eating obstructions and animal suffering and death [21]. Beyond animal suffering, these viruses cause worldwide economic losses by depressing meat and milk production as well as cattle by-products [10,11].

Recently, molecular biology techniques have allowed for the identification and characterization of several new and putative new types (PNTs) of BPV in bovine skin papilloma lesions in the Western Amazon region and in other regions of Brazil [5,6,12]. Herein, we aimed to genetically and histologically characterize papillomatous lesions from the upper gastrointestinal tract of beef cattle raised extensively in the Western Amazon region, Brazil, providing data for future epidemiological studies of cattle GIT lesion etiologies.

## 2. Material and Methods

### 2.1. Research Ethics

The approval by the Ethics Committee in the Use of Animals is not required once the specimens were collected from cattle slaughtered in slaughterhouses registered by the Brazilian sanitary inspection (Brazilian Law No 11794/2008). All rumen samples were collected by a veterinarian.

### 2.2. Sample Collection

All gross neoplastic lesions were collected from 100 animals, totaling 100 samples of papillomatous lesions from the bovine rumen (one sample per animal) (Figure 1) in slaughterhouses in the central region of Rondônia state, Northern Brazil. The samples came from two slaughterhouses in Ji-Paraná and Urupá municipalities, which receive cattle from Ji-Paraná, Alvorada do Oeste, Urupá, Teixeirópolis and Mirante da Serra municipalities, Brazil. The sampling was performed using a sterile scalpel and tweezers for each lesion. Subsequently, half of each sample was conditioned in 10% formalin (according to tissue availability) and the other half was refrigerated and stored at −20 °C.

### 2.3. PCR, Sequencing and Sequence Analysis

DNA extraction was performed using the commercial Purelink^®^ Genomic DNA Mini kit (Invitrogen, Carlsbad, CA, USA), according to the manufacturer’s instructions. FAP59/FAP64 primer pairs were used for partial amplification of the L1 gene [28]. The PCR reactions were performed with 2 µL of the extracted DNA, 0.2 μM of each primer, 1 unit of *Taq* DNA polymerase (Invitrogen, Carlsbad, CA, USA), 2.5 μL of 10× PCR buffer, 0.38 mM MgCl_2_, 0.05 mM of each dNTP and sterile ultrapure water, to the final volume of 25 µL. Amplifications were performed in a thermocycler under the following time and temperature conditions: 5 min at 95 °C, followed by 40 cycles of 1 min at 94 °C, 1 min at 50 °C, 1 min at 72 °C and a final extension of 7 min at 72 °C. Afterwards, 5 µL aliquots of the amplification reactions were subjected to electrophoresis in a 2% agarose gel, using Gel Red (Quatro G Biotechnology, Porto Alegre, Brazil), and visualized in a transilluminator UV LTB HE (Loccus, Cotia, Brazil).

The PCR positive samples were purified using the commercial kit NucleoSpin Extract II (Macherey—Nagel, Duëren, Germany) and subsequently sequenced using the automatic sequencer ABI-PRISM 3100 Genetic Analyzer armed with 50 cm capillaries and POP6 polymer (Applied Biosystems, Waltham, MA, USA), with forward and reverse primers. The sequences were edited using Geneious Prime software (version 2023.1.2). BLASTn tool (http://www.ncbi.nlm.nih.gov/BLAST, accessed on 5 April 2024) was used to compare the identity of the sequences obtained in this study with the sequences deposited in public databases (GenBank).

### 2.4. Phylogenetic Analysis

All BPVs reference genomes, as well as the sequences most similar to those obtained in this study, were retrieved from the NCBI (https://www.ncbi.nlm.nih.gov) for phylogenetic analysis. The alignment was performed with Clustal W [6,28,29] and the phylogenetic tree was built with the maximum likelihood method (ML) with the most suitable nucleotide substitution model, according to the “Find Best DNA/Protein Model” tool available in MEGA X (version 10.2.6) [30,31,32]. The reliability of the tree was tested with 1000 nonparametric bootstrap analyses.

### 2.5. Histopathological Analysis

Samples of papillomatous lesions from 100 animals were fixed in 10% buffered formalin and routinely processed for histology. For this, the samples were trimmed, dehydrated in increasing concentrations of alcohol, clarified in xylene solution and embedded in paraffin wax. Tissue sections (3–5 μm) were stained with hematoxylin and eosin (HE) and evaluated under light microscopy. Based on histological findings, the tumors were classified as fibropapilloma or squamous papilloma [33].

## 3. Results

### 3.1. PCR, Genetic Sequencing and Phylogenetic Analysis

For the 100 rumen samples subjected to PCR with the FAP 59/64 primer pair, papillomavirus (PV) DNA was amplified in 41 samples and not amplified in 59 samples using the well-established PCR with FAP59/FAP64 primer pair (Table 1 and Table 2). Phylogenetic analysis was performed for 22 samples that presented high-quality sanger sequencing (electropherogram showing a single peak to a single nucleotide). In the generated phylogenetic tree, the vast majority of sequences clustered in the genus *Deltapapillomavirus* (19 sequences) and three sequences grouped in an unclassified genus (Figure 2A,B), in the BPV44 cluster.

The samples belonging to genus *Delta* PV were classified as BPV2 (40.91%; 9/22) and BPV13 (45.45%; 10/22). Samples classified as BPV2 showed a high degree of identity with each other (100%) and with the reference BPV2 (98.02%). The sequences that clustered with BPV13 (100% identity) also showed 100% identity with each other.

The 21RO19R and 38RO19R sequences (100% similarity between them) have a common ancestor, namely BPV44. Greater phylogenetic distance was observed between the study sequence 49RO19R and BPV44 (only 94.42% of similarity), which can be classified as a possible new viral subtype. Results are shown in Figure 2 and Table 1, Table 2 and Table 3. The average age and weight of the slaughtered animals sampled were 33 months-old and 480 kg, respectively.

### 3.2. Histopathological Analysis

Histopathological analysis was performed on 22 samples, which had enough material for this purpose. The results show that nineteen (79.10%) can be histologically classified as fibropapilloma and two as squamous papilloma. Additionally, two samples (8.4%) did not show any histopathological alteration and one sample (4.10%) was inconclusive (Table 3). Fibropapillomas were characterized by mild mucosal hyperplasia, which had a wavy surface and formed long epithelial pins (rete pegs) toward the proliferated submucosa (Figure 3A).

The cell proliferation in submucosa was fusiform and formed bundles arranged in random flows, supported by a mild-to-moderate myxoid or collagenous stroma (Figure 3B). There was moderate anisocytosis and anisokaryosis and no mitotic activity. Squamous papilloma consisted of marked mucosal hyperplasia and ortho or parakeratotic hyperkeratosis, with a finger-like or slightly flat surface (Figure 3C). Keratinocytes often had eccentric pyknotic nuclei and a perinuclear halo (consistent with koilocytes). Enlarged keratohyaline granules and intranuclear amphophilic inclusions (Figure 3D) were observed in only two cases.

## 4. Discussion

Herein, it was possible to amplify PV sequences in 41/100 rumen papillomatous lesions using primer pair FAP59/64. Moreover, we report the occurrence of BPV44 in cattle rumen, which was first identified in bovine teat papilloma lesions using rolling circle amplification followed by high-throughput sequencing [6].

From the 59 negative papillomatous samples, 22 were histologically diagnosed as squamous papilloma or fibropapillomas. This fact may be due to the fact that the FAP primer pair is not able to amplify some types of BPV that were already detected in this region due to the lower affinity for the primers [34,35].

The primer pair FAP59/64 allowed for the amplification of BPV mostly from the *Deltapapillomavirus* genus (BPV2 and 13). The FAP primer pair has been widely and satisfactorily used in previous BPV screening studies [17,32]. It is important to point that these primers were originally designed to amplify human PVs and have been widely used to amplify PV from a vast range of species worldwide, which made it possible to describe several new PV types from distinct animal species [1,5,12,17,28,29,30,31,32,33,34,35].

Furthermore, two samples were classified as BPV44 and one as a possible new subtype of this recently described viral type. BPV44 was previously isolated in cattle teat lesions in southern Brazil [6], but we describe here the first detection of this viral type in ruminal lesions.

The histological findings of most samples in the current study are consistent with those typically seen in bovine cutaneous papilloma lesions [17,27,36,37]. BPVs of the *Deltapapillomavirus* genus (BPV1, 2, 13 and 14) usually induce the formation of fibropapilloma, as they infect the epidermis and dermis [5]. In this study, BPV2 and 13 were detected in papillomatous lesions mostly classified as fibropapilloma.

*Delta* PV have already been detected in papillomatous skin lesions of cattle in Rondônia state [12,17,34], in Southern Brazil [5,7,34,35,36,37,38] and in southeast Brazil [38,39]. Additionally, BPVs 1, 2 and 5 were detected in the mouth, esophagus, rumen and reticulum fibropapillomas of cattle and buffaloes in India, Japan and Brazil [21,39,40], corroborating to the findings of this study. *Delta* PVs are considered high-risk since they are frequently detected in neoplasms of the upper gastrointestinal tract and urinary bladder, especially when they are associated with the consumption of *Pteridium* species [7,8,12,41]. In the present study, it was not possible to obtain data on the farms. However, the region presents data on the incidence of the *Pteridium* species and can be a damaging factor; in addition, the biodiversity of other toxic plants may be related to the incidence [41].

Other studies detected BPV4 (*Xi*-PV) in GIT papillomatous lesions of cattle in the UK [22] and Italy [42,43]. On the other hand, members of the *Xipapillomavirus* genus, such as BPV12, were described in a cattle tongue in Japan [44], although these viral types were not detected in this study. BPV4, which has already been found in the GIT, is probably not a predominant viral type in northern Brazil, as it was not detected in any previous study carried out on cattle in the Amazon region, not even using high-throughput sequencing [17,34]. Other studies of GIT papillomas in bovines have already been associated with BPV1, 2, 4 and 5 [21,22,42]. However, regression may not occur in chronically immunodepressed animals and some types of PV have been linked to malignancy, especially in synergism with some chemical or environmental carcinogens [21]. These infections can extend from the mouth and tongue to the esophagus, rumen and reticulum and present low to high growth, with a nodular, spherical and pedunculated shape, which can result in difficulty in feeding and breathing, obstructive bloat and cause the death of the animal [21,22,23,44,45,46,47].

BPV1 DNA is commonly found in skin warts, bovine peripheral blood, placenta, amniotic fluid and bovine colostrum [12,14,19,39,48] and it has been suggested that the bloodstream contributes to viral dissemination [39,49,50,51,52]. Similarly, the identification of BPV2 and BPV13 in the present study could also be explained by bloodstream dissemination. Although the pathogenesis of the PVs involves epitheliotropism, its genetic material can be found in several different tissues in the same animal and also in the peripheral blood of their offspring [46,52].

## 5. Conclusions

Here, we were able to identify *Delta* papillomaviruses and the recently identified BPV44 infecting the rumen of cattle from the western Amazon, Brazil. This study shows that ruminal papillomatosis is a post-mortem finding, emphasizing the importance of this type of study for a better understanding of the pathologies caused by BPV.

## Figures and Tables

**Figure 1 animals-14-02262-f001:**
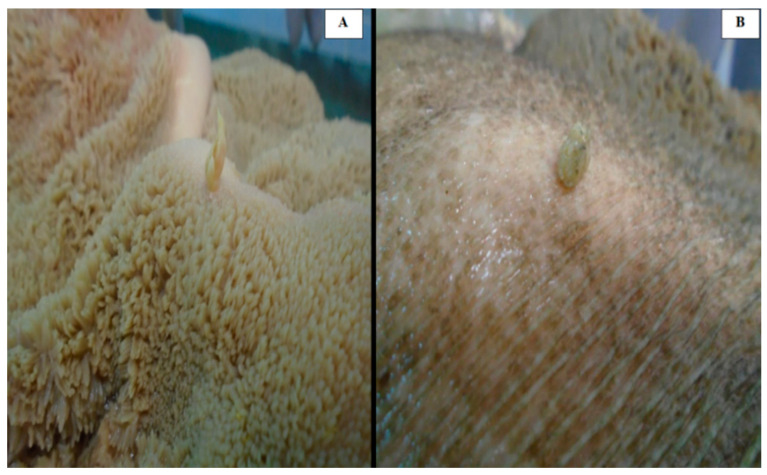
Papillomatous lesions found at bovine rumen. Exophytic lesions with filiform (**A**) and nodular (**B**) shape.

**Figure 2 animals-14-02262-f002:**
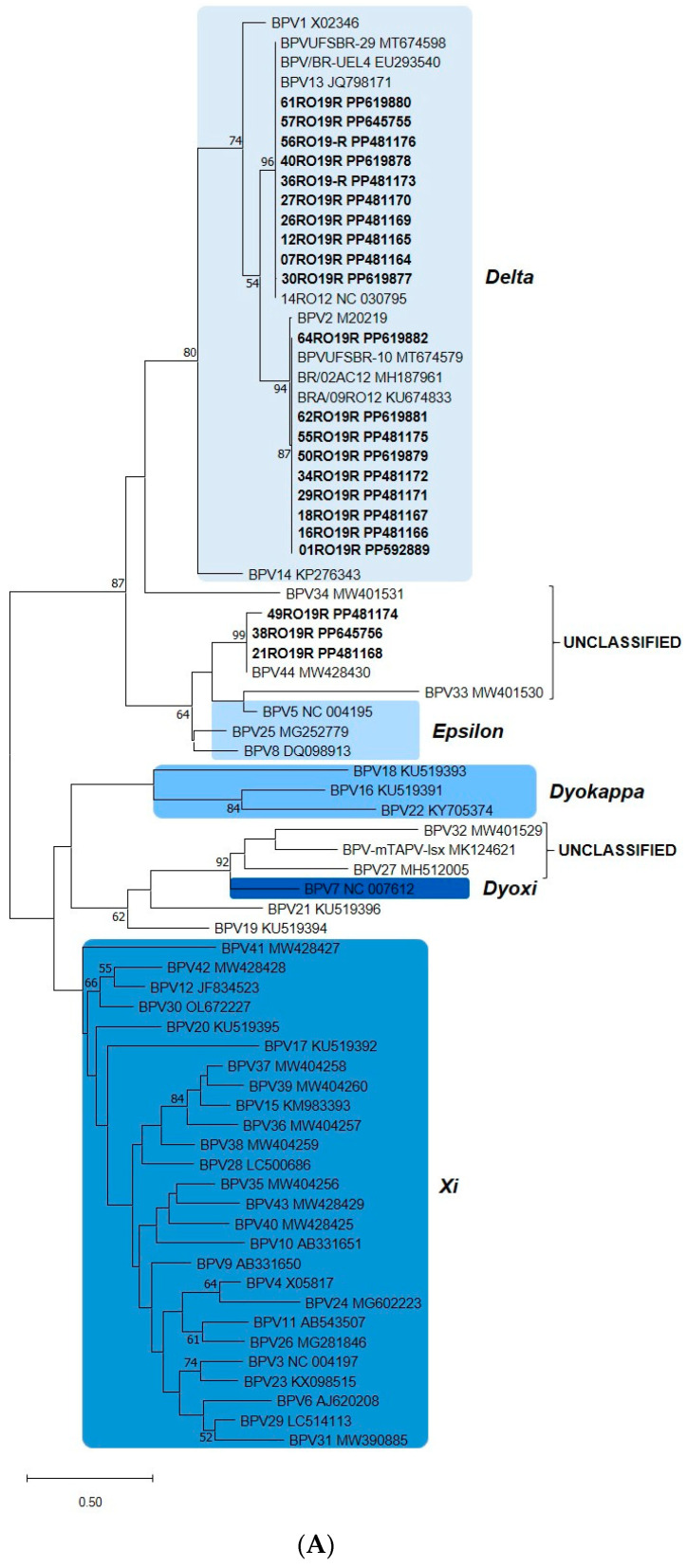
(**A**) Phylogenetic tree of *Bos taurus Papillomavirus* (BPV) sequences detected in the gastrointestinal tract (GIT). Dataset based on partial L1 sequences. The study samples and the most similar sequences, as well as the reference sequences of each type of BPV, were included in the analysis, totaling 73 sequences. Evolutionary analysis inferred by the maximum likelihood method, 3-parameter Tamura model, discrete Gamma distribution and bootstrap of 1000 replicates. The probability that taxa cluster together is shown next to the branches. Bootstraps <50% have been suppressed. Highlighted study sequences. (**B**) Nucleotide identity matrix between sequences generated by the alignment of partial L1 fragments of the BPVs of the study. The figure was generated in SDT software (version 1.2).

**Figure 3 animals-14-02262-f003:**
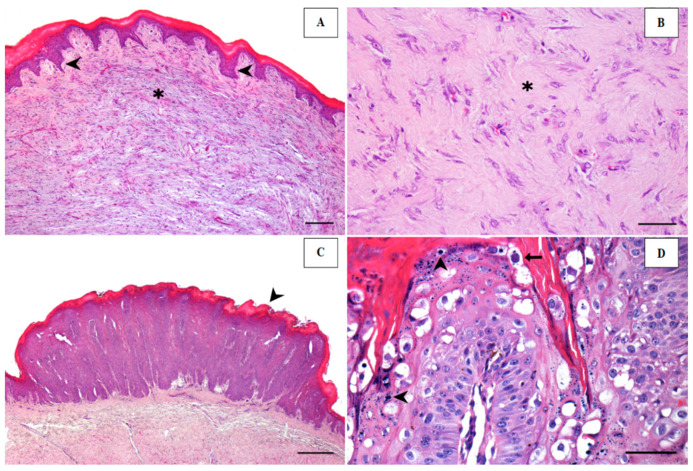
Papillomatous lesions from bovine rumen. (**A**) Fibropapilloma. Submucosal proliferation (asterisk) closely associated with mildly hyperplastic mucosa, which exhibits evident epidermal rete pegs (arrowheads). The mucosa is covered by orthokeratotic hyperkeratosis. Bar, 200 µm. (**B**) The proliferated cells of the submucosa show moderate pleomorphism and are supported by collagenous stroma (asterisk). Bar, 100 µm. (**C**) Squamous papilloma. Squamous mucosal epithelial proliferation with multiple exophytic projections and orthokeratotic hyperkeratosis (arrowhead). Bar, 200 µm. (**D**) Keratinocytes showing an increase in keratohyaline granules (arrowheads) or intranuclear amphophilic inclusion (arrow). Bar, 100 µm.

**Table 1 animals-14-02262-t001:** Negative samples in PCR and positive in histopathological diagnosis.

Municipalities *	Sex	Sample	Histopathological Diagnosis
**Alvorada do Oeste**	♂	02RO19R	Fibropapilloma
	♂	03RO19R	Fibropapilloma
	♂	04RO19R	Fibropapilloma
	♂	05RO19R	Fibropapilloma
	♂	06RO19R	Fibropapilloma
	♂	08RO19R	Fibropapilloma
	♂	10RO19R	Squamous papilloma
	♂	11RO19R	Squamous papilloma
	♂	14RO19R	Fibropapilloma
	♂	15RO19R	Fibropapilloma
	♂	17RO19R	Fibropapilloma
	♂	19RO19R	Fibropapilloma
	♂	20RO19R	Squamous papilloma
**Ji-Paraná**	♂	22RO19R	Squamous papilloma
	♂	28RO19R	Fibropapilloma
	♂	31RO19R	Squamous papilloma
	♀	35RO19R	Squamous papilloma
	♀	37RO19R	Fibropapilloma
**Mirante da Serra**	♀	39RO19R	Fibropapilloma
	♀	49RO19R	Squamous papilloma
	♀	51RO19R	Fibropapilloma
	♀	52RO19R	Fibropapilloma
	♀	53RO19R	Fibropapilloma
	♀	63RO19R	Squamous papilloma
	♀	65RO19R	Squamous papilloma
	♀	70RO19R	Squamous papilloma
**Urupá**	♀	71RO19R	Fibropapilloma
	♀	72RO19R	Fibropapilloma
	♀	73RO19R	Squamous papilloma and Fibropapilloma
	♀	74RO19R	Fibropapilloma
	♀	84RO19R	Fibropapilloma
	♀	85RO19R	Squamous papilloma
	♀	95RO19R	Fibropapilloma
	♀	98RO19R	Squamous papilloma

Subtitle: * Municipalities from Rondônia state, Brazil.

**Table 2 animals-14-02262-t002:** Negative samples in PCR and negative/inconclusive in the histopathological diagnosis.

Municipalities *	Sex	Sample	Histopathological Diagnosis
**Alvorada do Oeste**	♂	09RO19R	No Change
	♂	13RO19R	No Change
	♂	25RO19R	Inconclusive **
**Ji-Paraná**	♀	36RO19R	No Change
	♀	42RO19R	No Change
	♀	43RO19R	No Change
**Mirante da Serra**	♀	46RO19R	No Change
	♀	47RO19R	No Change
	♀	58RO19R	Non-diagnostic sample
	♀	66RO19R	No Change
	♀	67RO19R	No Change
	♀	68RO19R	No Change
	♀	69RO19R	Non-diagnostic sample
	♀	71RO19R	Inconclusive **
	♀	72RO19R	Inconclusive **
**Urupá**	♀	76RO19R	Inconclusive **
	♀	77RO19R	No Change
	♀	78RO19R	Inconclusive **
	♀	79RO19R	Inconclusive **
	♀	81RO19R	Inconclusive **
	♀	82RO19R	Inconclusive **
	♀	83RO19R	No Change
	♀	86RO19R	Inconclusive **
	♀	87RO19R	Inconclusive **
	♀	88RO19R	Inconclusive **
	♀	89RO19R	No Change
	♀	90RO19R	Inconclusive **
	♀	92RO19R	Non-diagnostic sample
	♀	94RO19R	No Change
	♀	99RO19R	Inconclusive **
	♀	100RO19R	Inconclusive **

Subtitle: * Municipalities from Rondônia state, Brazil; ** Inconclusive: Samples with histological lesions of mild acanthosis and hyperkeratosis.

**Table 3 animals-14-02262-t003:** Data on the sampled cattle, types of BPV and new putative types of BPV found in this study.

BPV Type	Genus	Municipalities *	Sex	Identity **	Sample	Histopathological Diagnosis	GenBank Access No.
**BPV2** **M20219**	*Delta*	Alvorada do Oeste	♂	98.02%	01RO19R	Fibropapilloma	PP592889
			♂	98.02%	16RO19R	Fibropapilloma	PP481166
			♂	98.02%	18RO19R	Squamous papilloma	PP481167
		Ji-Paraná	♀	98.02%	29RO19R	Inconclusive	PP481171
			♀	98.02%	34RO19R	Fibropapilloma	PP481172
			♀	98.02%	50RO19R	Fibropapilloma	PP481168
			♀	98.02%	55RO19R	Fibropapilloma	PP619879
			♀	98.02%	62RO19R	Fibropapilloma	PP619881
			♀	98.02%	64RO19R	Fibropapilloma	PP619882
**BPV 13** **JQ798171**	*Delta*	Alvorada do Oeste	♂	100%	07RO19R	Fibropapilloma	PP481164
			♂	100%	12RO19R	Fibropapilloma	PP481165
			♂	100%	26RO19R	Fibropapilloma	PP481169
		Ji-Paraná	♂	100%	27RO19R	Fibropapilloma	PP481170
			♂	100%	30RO19R	Fibropapilloma	PP619877
			♀	100%	36RO19R	No Change	PP481173
			♀	100%	40RO19R	Fibropapilloma	PP619878
		Mirante da Serra	♀	100%	56RO19R	Fibropapilloma	PP481176
			♀	100%	57RO19R	Fibropapilloma	PP645755
			♀	100%	61RO19R	Fibropapilloma	PP619880
**BPV44 MW543422**	Unclassified	Ji-Paraná	♂	100%	38RO19R	Squamous papilloma	PP645756
		Alvorada do Oeste	♂	100%	21RO19R	No Change	PP481168
				94.42%	49RO19R	Squamous papilloma	PP481174

Subtitle: * Municipalities from Rondônia state, Brazil; Sex: ♂ Male and ♀ Female. ** Based on reference BPV genome.

## Data Availability

Data are contained within the article and genetic sequences can be accessed by the GenBank accession numbers provided.

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
