# Peer review of "Characterization of Bovine Papillomavirus Types Detected in Cattle Rumen Tissues from Amazon Region, Brazil"

_animals, 2024, doi:10.3390/ani14152262_

Round 1

Reviewer 1 Report

Comments and Suggestions for Authors

This paper provides an analysis of the BPV species present in the rumen of cows slaughtered in the Western Amazon region of Brazil. All in all it is a well written and informative paper which provides new information about the distribution of BPV types in this region.

The research concept is clear, the writing language and references adequate.

I have two small notes/questions which I would like addressed:

1. In the Discussion the authors note the possible consequences of BPV infection for the infected cattle (line 227-235). I think the Introduction would also benefit from a few sentences about this topic and about the damage that this poses for the industry (if any). This could serve to emphasize the importance of these kinds of studies.

2. I am not entirely clear as to which samples were chosen for histology? Were these exclusively the samples that were negative in PCR analysis (lines 164 and 193)? If so, why? I understand that you chose the samples that had enough material to work with but was this the only criteria? I would appreciate if you could edit some of the text to clarify this.

3. I noticed that a few Back Matter sections are missing. Namely, the description of Supplementary figure 1, author contributions, funding and conflicts of interest. Also, you should include the data availability statement, if you uploaded the novel BPV44 variant sequences anywhere.

4. There is a portion of a sentence repeated in line 42.

All in all, this is an informative paper and I recommend publication after these minor matters are addressed.

Author Response

Reviewer 1

Comments and Suggestions for Authors

  1. In the Discussion the authors note the possible consequences of BPV infection for the infected cattle (line 227-235). I think the Introduction would also benefit from a few sentences about this topic and about the damage that this poses for the industry (if any). This could serve to emphasize the importance of these kinds of studies.

ANWSER: Thank you for the carefully revision. We agree to the suggestions and have added information to the Introduction section.

 “BPV infection of the TGI can extend from the mouth to tong, rumen, and reticulum and can present high mass growth, which can result to breathing and eating obstructions and animal suffering and death [21]. Beyond animal suffering, these viruses cause worldwide economic losses by depressing meat and milk production as well as cattle by-products [10, 11].”

  1. I am not entirely clear as to which samples were chosen for histology? Were these exclusively the samples that were negative in PCR analysis (lines 164 and 193)? If so, why? I understand that you chose the samples that had enough material to work with but was this the only criteria? I would appreciate if you could edit some of the text to clarify this.

ANSWER: All samples which had enough material (papilloma) for histology were analyzed. It comprised 85 samples.

  1. I noticed that a few Back Matter sections are missing. Namely, the description of Supplementary figure 1, author contributions, funding and conflicts of interest. Also, you should include the data availability statement, if you uploaded the novel BPV44 variant sequences anywhere.

ANSWER: Supplementary figure 1 are not present in this study and it was corrected in the text. Authors conflict of interest was added in the text as well as Authors Contributions. GenBank accession number of sequences related to BPV44 are available on table 3.

  1. There is a portion of a sentence repeated in line 42.

ANSWER: Thank you for the revision. The mistake was corrected.

Reviewer 2 Report

Comments and Suggestions for Authors

The manuscript entitled: "Characterization of Bovine Papillomavirus types infecting bovine rumen in the Amazon region, Brazil" presents an important study aimed at genetically characterizing Bovine Papillomavirus (BPV) present in rumen neoplastic lesions of extensively raised cattle in the Western Amazon region of Brazil. The study used PCR, sequencing, and phylogenetic analyses to identify BPV types, including BPVs 2, 13, and 44, and reports the identification of a new subtype of BPV44. I think this study could contribute to the improved understanding of bovine pappiloma virus epidemiology in the current study area. However, substantial revision is required to clarify the research focus, improve readability, and ensure technical accuracy.

General comments

·        Language use and writing: full of grammatical mistakes, spelling errors, and use of non-technical words, making it difficult to read the manuscript. A thorough proofreading and revision for language and style are required.

·        Some of the technical terms and abbreviations are used without being defined on first use (e.g., BPV should be defined at first mention in the abstract and main text and figures).

·        The manuscript lacks a clearly articulated research question or statement of the research gap. It would be great if the authors could try to define the specific aspect of BPV epidemiology the study addresses and how it builds on or diverges from existing research.

Some specific comments

For example, many misspelled words are in lines 38 and 70.

Title: Authors can improve the title to make it clear. For example: Characterization of bovine papillomaviruses detected in rumen tissues of cattle from the Amazon region, Brazil."

Abstract

·        Line 18: the sentence isn't complete. Does it refer to the difficulty of sample collection? Or..?

·        There are a few misspelled words throughout the manuscript. For example, "hytologically" should be corrected to "histologically" in the abstract.

Introduction

·        Line 38:is it leading?

  • Expand on the relevance of Bovine GIT papillomavirus in cattle, describing its impact and the study's contribution to existing knowledge.

Methods:

The methods section is generally well-described. However, the manuscript would benefit from more detailed descriptions of the sample collection process and the criteria used for sample selection. For example, no information on the cattle population type, breed, feed, farming condition, husbandry, etc., that could affect the outcome ( virus infection status of cattle).

Line 80:  sentence not clear

Line 84: this sentence seems a result, not a methods.

Line 105: Does this sentence refer to purifying the PCR product or sample? Please revise the sentence.

Line 11: Does it mean publicly available data from NCBI?

The manuscript lacks a detailed statistical analysis of the data, which could strengthen the conclusions. For example, statistical tests to compare the prevalence of different BPV types among different municipalities or age groups of cattle or any other variables.

Result and discussion

Line 131: in this context, what is high-quality sequencing?

What are the main findings of the current study? Especially in the discussion section, the main findings of the current study are not clearly indicated and discussed.

Comments on the Quality of English Language

Please refer to the attached report.

Author Response

Reviwer 2

General comments

  1. Language use and writing: full of grammatical mistakes, spelling errors, and use of non-technical words, making it difficult to read the manuscript. A thorough proofreading and revision for language and style are required.

ANSWER: The full text was send to English review as requested.

  1. Some of the technical terms and abbreviations are used without being defined on first use (e.g., BPV should be defined at first mention in the abstract and main text and figures).

ANSWER: The text was carefully revised and the abbreviations and technical terms were corrected.

  1. The manuscript lacks a clearly articulated research question or statement of the research gap. It would be great if the authors could try to define the specific aspect of BPV epidemiology the study addresses and how it builds on or diverges from existing research.

ANSWER: Our study aimed to characterize histologically and genetically the papillomatous lesions found at the gastrointestinal tract (GIT) of cattle raised extensively in Western Amazon Region. Additionally, the study provides data of GIT lesions etiologies (herein BPV). Regarding Bos taurus Papillomavirus lesions, the most common studies focuses at skin lesions followed by urinary bladder studies, consequently, studies focusing at GIT samples collected at slaughterhouses are scarce. Therefore, the present study is scientifically relevant to the research area.

Some specific comments

  1. For example, many misspelled words are in lines 38 and 70.

ANSWER: All misspelled were corrected.

  1. Title: Authors can improve the title to make it clear. For example: Characterization of bovine papillomaviruses detected in rumen tissues of cattle from the Amazon region, Brazil."

ANSWER: We agree and have modified the title to “Characterization of Bos taurus Papillomavirus types detected in cattle rumen tissues from Amazon region, Brazil”

Abstract

  1. Line 18: the sentence isn't complete. Does it refer to the difficulty of sample collection? Or..?

ANSWER: Thank you for the review, we agree and have modified the sentence.

“The Bos Taurus Papillomavirus (BPV) can cause lesions in the mucosa of the gastrointestinal tract (GIT) of cattle and induce the formation of papillomas in organs such as the pharynx, esophagus, rumen and reticulum. GIT papillomas can leading to feeding and breathing distress as well as samples collection difficulty which reduces the BPV diagnosis in these organs.”

  1.  There are a few misspelled words throughout the manuscript. For example, "hytologically" should be corrected to "histologically" in the abstract.

ANSWER: Thank you for the review, the misspelled was corrected.

Introduction

  1.   Line 38:is it leading?

ANSWER: Thank you for the review, the misspelled was corrected to “...can lead to…”.

  1. Expand on the relevance of Bovine GIT papillomavirus in cattle, describing its impact and the study's contribution to existing knowledge

ANSWER: BPV infection of the TGI can extend from the mouth to tong, rumen, reticulum and present high mass growth, which can result to breathing and eating obstructions and animal suffering and death [1]. Beyond animal suffering, these viruses cause worldwide economic losses by depressing meat and milk production as well as cattle by-products [2, 3]. Considering this information, it is important to understand the viral types that occurs in cattle, contributing to future epidemiological studies and vaccine studies.

Methods:

  1. The methods section is generally well-described. However, the manuscript would benefit from more detailed descriptions of the sample collection process and the criteria used for sample selection. For example, no information on the cattle population type, breed, feed, farming condition, husbandry, etc., that could affect the outcome ( virus infection status of cattle).

ANSWER: Criteria used for sample collection was added as requested.

“All gross neoplastic lesion was collected from 100 animals, totaling 100 samples of papillomatous lesions from the bovine rumen…”

Regarding cattle information requested, unfortunately, as it is very complex to have industry access, especially inside the slaughterhouses it was not authorized access data collection from any animal sampled.

  1. Line 80:  sentence not clear

ANSWER: Sentence was modified to make it clear.

“All gross neoplastic lesion was collected from 100 animals, totaling 100 samples of papillomatous lesions from the bovine rumen (one sample per animal)…”

  1. Line 84: this sentence seems a result, not a methods.

ANSWER: Sentence was removed from Material and methods section and added to to the final session of Results section.

  1. Line 105: Does this sentence refer to purifying the PCR product or sample? Please revise the sentence.

ANSWER: Sentence was modified to better understanding.

“The PCR positive samples…”

  1. Line 112 : Does it mean publicly available data from NCBI?

ANSWER: GenBank is the National Library of Medicine for the National Center for Biotechnology Information, which collect and make available genetic data from other DNA data centers.

  1. The manuscript lacks a detailed statistical analysis of the data, which could strengthen the conclusions. For example, statistical tests to compare the prevalence of different BPV types among different municipalities or age groups of cattle or any other variables.

ANSWER: Unfortunately the slaughterhouses did not authorized any other data access beyond gender and general weight and age from sampled animals. Therefore, statistical tests were not the study focus. Additionally, similar studies of TGI BPV occurrence are scarce and do not provide statistics analysis.

Result and discussion

  1. Line 131: in this context, what is high-quality sequencing?

ANSWER: Sequences that not presented double peak and noises in the first sequencing analysis in order to assemble the consensus sequences, which were used to phylogenetic tree.

  1. What are the main findings of the current study? Especially in the discussion section, the main findings of the current study are not clearly indicated and discussed.

ANSWER: Thank you for the thoughtful review. We have modified the discussion to make the results clear.

“Herein we could amplify 41/100 rumen papilomatous lesions by using primer pair FAP59/64 and report the occurrence of BPV44 in cattle rumen, which was first diagnosed in bovine teat papilloma lesions using rolling circle amplification followed by high-throughput sequencing...”

References

  1. Yamashita-Kawanishi, N., Tsuzuki, M., Kasuya, F., Chang, H. W., Haga, T., Genomic characterization of a novel bovine papillomavirus type 28, Virus Genes 56 (2020) 594–599, https://doi.org/10.1007/s11262-020-01779-9.
  2. Yamashita-Kawanishi, N., ITO, S., Ishiyama, D., Chambers, J. K., Uchida, K., Kasuya, F., Haga, T., Characterization of Bovine papillomavirus 28 (BPV28) and a novel genotype BPV29 associated with vulval papillomas in cattle, Veterinary Microbiology 250 (2020) 108-879, https://doi.org/10.1016/j.vetmic.2020.108879
  3. Kumar, P., Nagarajan, N., Saikumar, G., Arya, S. R., Somvanshi, R., Detection of bovine papilloma viruses in wart-like lesions of upper gastrointestinal tract of cattle and buffaloes. Transboundary and Emerging Diseases 62 (2015) 264–271, https://doi.org/10.1111/tbed.12127.

Round 2

Reviewer 2 Report

Comments and Suggestions for Authors

I think the term “Bos taurus Papillomavirus” isn’t commonly used. I suggest you use bovine Bos taurus Papillomavirus instead. For more information on virus name and taxonomy, you can refer to (Papillomaviridae, Taxonomy ID: 10571) of NCBI: https://www.ncbi.nlm.nih.gov/labs/virus/vssi/#/virus?SeqType_s=Nucleotide&VirusLineage_ss=Bovine%20papillomavirus,%20taxid:10571&utm_source=nuccore&utm_medium=referral

 Please change the” Bos taurus Papillomavirus” to Bovine Papillomavirus.

1.        There are still language use issues with the manuscript. For example:

·        In the abstract, the sentence: “The present study could identify 38 BPV types never described in papillomatous samples from cattle rumen of Brazilian Amazon region, 39 including BPV13 and BPV44.” It doesn’t seem correct.

·        In line 74, the term “tong” is used, which isn’t part of the GIT of cattle.

·        Line 93: In some cases, capital and small cases are used incorrectly.

·        Line 2005:  the sentence “Herein we could amplify PV sequences in 41/100 rumen papilomatous lesions using 205 primer pair FAP59/64” should be revised to indicate what has already been done.

·        Line 215: in “bovine BPV”, the term bovine is redundant.

·        Line 240: tonge or tongue?

The paragraph in lines 66 to 77 should be revised to avoid repetitions.

2.        Authors should try to present the manuscript clearly. For example, rather than writing  “high-quality sequencing” in the results section, use sequence quality measurement matrices to indicate that the sequences generated were high quality.

3.        Line 166: what is “13 (45.45%; 10/22)” refereeing, or is it BPV13 (45.45%; 10/22)? The sentence is confusing.

4.        Line 262: no discussion part describes the “underdiagnosed” condition of BPV. Either you discuss it and then conclude or remove it from the conclusion. 

Comments on the Quality of English Language

The manuscript has some spelling, consistency, and grammatical errors that need to be fixed.

Author Response

Dear, follow the corrections
